# The Ethical Code of Conduct for Physiotherapists—An Axiological Analysis

**DOI:** 10.3390/ijerph20021362

**Published:** 2023-01-12

**Authors:** Krzysztof Pezdek, Robert Dobrowolski

**Affiliations:** Department of Physical Education and Sport, Wroclaw University of Health and Sport Sciences, 51-612 Wroclaw, Poland

**Keywords:** ethical rules, values, axiological analysis, physiotherapy, physiotherapist

## Abstract

The goal of the article is an axiological analysis of the Ethical Code of Conduct for Physiotherapists. The basic ethical values constituting the axiological basis of physiotherapy are care, professionalism, responsibility, fairness, professional integrity, respect for a patient/client’s dignity and autonomy. Those values have been selected from the theory and practice of physiotherapy, but also from socio-cultural conditions influencing the relations and interdependencies between physiotherapists and other professional groups or society as a whole. Those values can exist as qualities of a subject (a physiotherapist) or as functions realised by them (acting for the welfare of a patient/client, society, profession). Some of the analysed values have been directly enumerated in the Ethical Code of Conduct for Physiotherapists, while others must be deduced from the rules included in this document. The analysed values should be internalised by the physiotherapists during their training and professional practice.

## 1. Introduction

The profession of a physiotherapist in Poland has been regulated by a special act since 25 September 2015. In 2022, in the Resolution no.: 25/II KZF/2022, the Ethical Code of Conduct for Physiotherapists in Poland was adopted [1], a result of the work of members of the Group on Ethics, operating by the Polish Chamber of the Physiotherapists [Krajowa Izba Fizjoterapeutów—KIF] since 2019 [2]. Those rules represent a system of values that is common for professionally active physiotherapists and constitutes a signpost both in individual conduct and in building the prestige of the job in society. Despite those legal regulations, words such as a masseur are still used to describe physiotherapists in social awareness. Obviously from the perspective of the current law such names are incorrect and reflect a low level of knowledge on the job in society [3,4].

Although the Code of Ethics of a Physiotherapist of the Republic of Poland, created by researchers associated with the Polish Society of Physiotherapy Polskie Towarzystwo Fizjoterapii—PTF] [5] existed before 2015, the document did not cover all professionally active physiotherapists, despite being fairly well accepted in the milieu and pointing the general direction of ethical thinking in the job. For organisational reasons, the document could not be discussed in the whole milieu and hence it did not constitute the conclusion of disputes on its content and structure. For this reason, the code was only a projection of certain attitudes and behaviours, desirable to the narrow group of physiotherapists, and it did not fully reflect the challenges the Polish physiotherapists faced. In that time, it was also not possible to exchange freely and fully the experience and knowledge between Polish physiotherapists and international organisations, since it was only in 2015 that a central Polish professional association (the Polish Chamber of the Physiotherapists) was created, gathering and representing all active members of the profession. Thanks to that, Polish physiotherapists could assemble and start partner relationships with other associations, such as the American Physical Therapy Association (APTA) or the Australian Physiotherapy Association (APA).

The axiological analysis undertaken in the present article aims at pinpointing the carriers of certain values and indicating certain functions that should be realised by a subject being a carrier of those values [6]. In the case of our analyses, the carriers of values are certain rules of the professional code of ethics of a therapist. Those values have been selected from the theory and practice of physiotherapy but also from socio-cultural conditions influencing relations and interdependencies between physiotherapists and different professional groups as well as society as a whole. Those basic values contained in the Ethical Code of Conduct for Physiotherapists include care, professionalism, responsibility, fairness, professional integrity, respect for the dignity and autonomy of a patient/client. Those values should be internalised by the physiotherapists during their training and professional practice, so that thanks to the values, the physiotherapists can act for the benefit of a patient/client, society and the profession (the function of values). Therefore, there are values understood as qualities of a subject (a physiotherapist) or as functions realised through them (a physiotherapist acting for the benefit of a patient/client, society and the profession). Some of the values are directly mentioned in the Ethical Code of Conduct for Physiotherapists, while others need to be deduced from the rules contained in that document.

## 2. Values and Ideals in a Physiotherapist’s Work

The values enumerated above, constituting the axiological basis of the profession of a therapist, at the same time create its ontical status. This status is expressed through ideals having both the incentive to undertake the profession and the signpost in the professional practice. The crucial professional ideal for physiotherapists is undoubtedly a person’s ability. This ideal complements other ideals of medical professions, which are, first of all, life and health.

Physiotherapists are obliged to focus on a person’s ability in their work, while at the same time caring for dignity and autonomy, being responsible and fair toward the distribution of their services and equalising opportunities. Their professional experience should be gained through the application of the current scientific knowledge and practice that is accepted in the physiotherapist milieu. All those activities should reinforce the integrity of physiotherapy among other medical professions in such a way that competence conflicts can be avoided and the distinctiveness and independence of physiotherapy as a field of knowledge can be stressed [3]. Realising the professional ideals, a physiotherapist can not only act therapeutically but also be an educator on healthy lifestyle, carry out research or teach, or run a business. All those functions have been provided for in the Ethical Code of Conduct for Physiotherapists.

## 3. Patient and Client in a Physiotherapist’s Practice

In the code of professional ethics adopted during II National Physiotherapists Reunion, there is a novelty: not only are patients mentioned as recipients of a physiotherapist’s services, but also clients.

Generally, the ethical aspect of relations between a therapist and a person undergoing therapy is realised in three basic configurations. The first of them (i) *a therapist–a patient*, is being increasingly often questioned in liberal cultures due to its authoritarian nature and whenever possible, a patient is not only granted the right to the conscious participation in the therapeutic process but also endowed with a power to control that therapeutic process to a certain degree. As a result, the traditionally hierarchical social relationship has been transformed into an egalitarian relation (ii) *a therapist–a client* [7,8]. The third type of a relation: (iii) *an analyst–an analysand* is, admittedly, the most rarely occurring one and technically happens most often within the frames of its primary environment, i.e., modern psychoanalysis, especially in the Lacanian circle. It is, however, worth mentioning, as it constitutes a peculiar intermediate state between the two types mentioned before: (i) *a therapist–a patient* and (ii) *a therapist–a client*. The crucial part of the model of therapeutic relation created by Jaques Lacan is the ethical duty of pursuing the truth. This imperative is so categorical that both the suffering of an obedient patient and the claims of a demanding client can become secondary to the pursuit of even the most painful truth. In this case, both sides of the relation are searching for an efficient interpretation; the main burden of that search lies with an analysand, while an analyst strives to remain an impartial helper. That relation is different from both the condescending attitude of a therapist to the passive and submissive patient and the contract between an expert and a client demanding the performance of a commissioned task [9,10].

Therefore, the change in the attitude towards the person undergoing the therapy is the answer to modern trends not only in physiotherapy. According to these trends, therapeutic relation abandons the stigmatising notion of *a patient* (a sick person, with disabilities, needing care) for the sake of the notion that stresses partner relations, which is *a client*. The change also reflects changes in the professional practice of physiotherapists in Poland. Not all professional functions are realised in health centres, where a physiotherapist deals with patients. In schools, at universities, in aquaparks, spa, fitness or wellness clubs, a physiotherapist does not work with patients but with fully healthy and able-bodied people, expecting advice on health prevention or life quality. In many documents regulating the ethics of a physiotherapist across the world, the word patient is no longer used, also in a therapeutic relationship, being substituted by the word *client* [11].

## 4. Values in the Code of Ethics of a Physiotherapist

### 4.1. Care

The basic value, constituting the axiological basis in the job of a physiotherapist, is care. The analysed document mentions that value three times (§1.1; §2.12–13). In §1.1, care is enumerated among the basic values constituting the axiological basis of the job. It is worth observing that care is listed as the first one. It is not a random decision; without care, moral regulation of physiotherapeutic services provided in current the socio-cultural conditions, would be impossible. In §2.12–13, care is mentioned with relation to touch, which is the basic tool of a physiotherapist. Touch, however, is a sense that can have many meanings depending on the context in which it is used, e.g., therapeutic, erotic, relaxing or abusive meanings [12,13]. A caring physiotherapist should be aware of the extra-therapeutic meaning of touch and respect the emotional, psychic and physical boundaries of a patient/client while he or she does their job. Additionally, a physiotherapist should be able to receive non-verbal messages sent by the bodies of persons he or she works with.

The exceptionality of care in physiotherapy consists in the fact that it marks the boundaries of relations between the professionals and patients/clients. Persons entering into relation with a physiotherapist do that in the spirit of care for themselves. They have knowledge/conviction that their quality of life is lower due to the discomfort in the physical sphere they feel. It does not need to be a concrete illness or disability but may as well be a conviction of decreasing ability caused by ageing, bad diet or a sedentary lifestyle. Starting this relation is therefore the result of a patient/client’s care for his or her health, quality of life or physical ability. From this point of view, care creates a bridge between the world of private experience of a patient/client and the healthcare system, represented by a physiotherapist. A relation based on care embraces both therapeutic activities and advisory or educational ones. Hence, a caring physiotherapist not only tries to provide therapeutic help to a given person, but also tries to change their bad habits concerning physical activity, diet, hygiene, body stance, etc. [14,15,16].

Attentive care for a patient/client requires a contextual approach on the part of a physiotherapist. It means that in providing physiotherapeutic care not only substantive activities, directly connected with the realised professional function, are taken into account, but also non-substantive ones, connected with a patient/client’s private life. Hence, physiotherapists are often confidants of their patient/client’s most intimate confessions which they might use in the professional relationship [16]. Such information, however, might make a physiotherapist dependent on the person whom he helps [17].

Physiotherapists working with patients with limited consciousness or unconscious are in a still different situation, as their patients are unable to verbalise their needs. Attentive care in such a case consists mainly in reading non-verbal signals (facial expressions, gestures, movements of extremities, sounds) and reacting to them with order to minimise discomfort (pain, suffering) [9].

### 4.2. Professionalism

Although the Ethical Code of Conduct for Physiotherapists mentions professionalism only once (§1.1), as early as in the introduction to that document, a physiotherapy is defined as a “self-standing medical profession”, while a physiotherapist is defined as a professional who “believes in, promotes and realises in his or her activities certain ideals, values and rules of the code of professional ethics” [1].

A professional physiotherapist acts in such a way as not to treat the patient/client instrumentally, as a means to achieve a certain goal (e.g., financial profits, professional promotion, ability to run a business, etc.), but to treat them autotelically, as a goal in itself. Such definition of professional activity places a physiotherapist in a position of service towards the good of a patient/client, for whom a professional should be ready to sacrifice even his or her own good. Hence, a physiotherapist is not only a properly qualified person from the technical side of the profession, regularly updating their knowledge and professional experience, but also a person capable of reflective assessment of their own professional status. A physiotherapist should do that from an axiological perspective in which ability, health and life occupy the highest place in the hierarchy of values.

Concrete rules regulating the relation between a physiotherapist and his or her profession have been enumerated in detail in §6.1–10 of the analysed document. The paragraph stresses first of all the significance of attitudes and behaviours that strengthen professional prestige and trust towards patients/clients, co-workers and society.

### 4.3. Responsibility

Responsibility has been mentioned in the Ethical Code of Conduct for Physiotherapists three times (§1.1; §7.1; §10.3). For the first time, it is mentioned in §1.1, among the values constituting the axiological basis of the profession. In §7.1, we have references bearing responsibility for publicly passed information, both in direct contact and via the Internet, social media or a communicator. A distinguishing feature of bearing responsibility is the lack of a direct link between the performer on an action and its consequences. It follows that a physiotherapist who publicly conveys information loses the control over it at some moment. It might be said that information starts to live its own life and the person who revealed it has no influence on the way it will be received by other people. For this reason, physiotherapists should not only protect the personal details of their patients/clients according to the current law, but also disseminate substantive content in such a way that they may be sure that they did everything on their part so as not to mislead their potential recipients, to ensure that information is in accordance with the facts and does not put anybody at risk intentionally. A physiotherapist should therefore exercise wisdom and reason and should not only accurately analyse what could be disseminated and in what form, but also what the consequences could be for private persons and representatives of different institutions.

Responsibility is mentioned for the third time in §10.3. That paragraph is entirely devoted to the rules of ethics in scientific research and is in accordance with the Declaration of Helsinki [18] and the Code of Ethics of a Researcher, adopted at the General Meeting of the Polish Academy of Sciences [19]. Responsibility in this approach consists in bearing responsibility by a physiotherapist for the choice of the objective of the research, the choice of an appropriate group of respondents, obtaining a conscious consent of the respondents (or their legal guardians), protecting the obtained data and obtaining a consent of a relevant ethical committee.

Physiotherapists on each stage of their research should undertake formal responsibility (pursuant to the binding law or regulations) and substitutive (for the third persons engaged in the research). They should also express readiness to be held responsible for the undertaken activities and their consequences. All above, mentioned situations connected with responsibility are linked by acting responsibly [20]. In the literature of the subject, it is often called moral responsibility [21]. Moral physiotherapists, independently of the value of the conducted research, financial means, researchers’ engagement, etc., always put the welfare of the patient/client first. Hence, they do everything to protect a patient/client from harm and to enable the patient/client’s resignation from taking part in research at any moment.

In undertaking responsibility and acting in a responsible way, there is a direct dependence between the performer of the action and its consequences. Hence, physiotherapists have control over realising certain tasks, and whether they perform the task well or badly, depends mostly on their substantive and organisational preparation and the experience they gained. In being held responsible, there does not need to be such a direct link between the action and its consequences, because in this case the assessment of the action or its consequences is based on the gathered evidence. Therefore, if physiotherapists in their actions focus on the merits of the case and scrupulously document these actions, and then archive the documents properly, then even in a problematic situation they can justify their reasons [3,20,21].

The situations of responsibility described above concern all the ethical rules of the profession of a physiotherapist, and not just the ones enumerated in §7 and §10, as responsibility can be deduced from other rules in their relevant meaning and then applied to a concrete professional situation.

### 4.4. Fairness

In the Ethical Code of Conduct for Physiotherapists, fairness has been mentioned twice (§1.1 i §10.3). In §1.1, it has been mentioned among the values constituting the axiological basis of the profession. Paragraph §10.3, on the other hand, deals with fairness in scientific research. It refers to the choice of the respondent sample, which should not be random but substantively justified. Additionally, each participant should express conscious and autonomous consent to take part in the research, as well as have guaranteed possibility to resign form the research at its each stage.

Apart from employing fairness in ethical rules governing the conduct of a physiotherapist in scientific research, the meaning of this value is constitutive for the whole profession, as fairness is the value that defines the boundaries of safe and acceptable performance of physiotherapeutic services. Hence, a physiotherapist should be aware that exceeding his professional competences might be assessed negatively, as acting in an unfair way. In particular, fairness interacts with care and responsibility. Therefore, a physiotherapist should care for a patient/client realising the rule of fair distribution of services. It means that he or she takes into account only the functional status of a patient/client and substantive, organisational and equipment-related possibilities of a performed service. From this point of view, there is no chance of any activities of a physiotherapist evidencing discrimination or unequal treatment, e.g., for the sake of race, sex, age, financial status, religion, views or sexual orientation.

Being directed in providing physiotherapeutic services by criteria from outside the field of physiotherapy is not only unfair, but also irresponsible, as both values (fairness and responsibility) complement each other. A fair person must be responsible at the same time, and a responsible person must be fair as well [3,21,22]. Therefore, caring and responsible performance of physiotherapeutic services should be in accordance with the rule of fairness that rationalises and objectifies relations between a physiotherapist and a patient/client, co-workers or society.

### 4.5. Professional Integrity

Professional integrity in the analysed document has been mentioned once, among the values constituting the axiological basis of the profession (§1.1). However, §4 and §11 are entirely devoted to these values, as they refer to the relations among co-workers, team members and the Polish Chamber of Physiotherapists authorities. From the axiological point of view, those relations should be, first of all, fair and responsible, based on the respect for dignity and autonomy. Physiotherapists should act in a way that strengthens professional integrity both within the profession and outside it. They should support their co-workers, express only substantive (not personal) opinions about them and resolve controversial issues in an official way. Additionally, physiotherapists should react to all behaviours of their colleagues that exceed their competences, are not substantive or attack the dignity and autonomy of other people. Such conduct may not only be harmful to certain people, but it also may have a detrimental effect on the profession, lowering the professional and social status connected with it.

A special responsibility rests, therefore, on physiotherapists belonging to multidisciplinary teams. Working in such teams, they care not only for the good of a patient/client, but also to preserve the integrity of their own profession so as to avoid a competence-related chaos in their relations with other members of the team, representing many different professions, also not medical, such as a psychologist, a lawyer or an occupational therapist [3].

Additionally, a physiotherapist should support the authorities in the Polish Chamber of the Physiotherapists in their efforts to improve working conditions and to ensure an increasingly positive reception of the profession in society.

### 4.6. Respect

In the Ethical Code of Conduct for Physiotherapists, respect is mentioned together with dignity and autonomy (§1.1). Nevertheless, each of these values works individually to create the axiological basis of the analysed profession. In ethical rules, respect has mostly instrumental significance. Therefore, it is used to achieve other values, such as professional integrity, fairness or responsibility. Treating somebody with respect is not difficult as long as he or she professes the same values as we do. Thus, a physiotherapist respects a client who has a similar attitude to the issues of health and therapy. He or she also respects a colleague who has a similar view on the adopted therapeutic method.

However, when there is a discrepancy between the systems of values of people entering into relations, one has to overcome one’s antipathy towards or even fear of an attitude, which one does not accept, and make an effort to put up with it. Such behaviour is defined as tolerance, and its boundaries are defined by respect. Therefore, when physiotherapists tolerate people professing other views than they hold, confessing another religion, listening to other music, supporting another club, etc. they behave in a fair way, responsibly and professionally. In a situation when the respect for diversity of beliefs, attitudes and conduct ends, tolerance ends as well, and oppression, discrimination, unequal treatment or even violence begin. It should also be remembered that the boundaries of tolerance are defined by social norms (formal and informal). Therefore, a behaviour that breaks such norms cannot be tolerated, e.g., inciting to committing a crime or to violence.

There are numerous fragments where the Ethical Code of Conduct for Physiotherapists encourages physiotherapists to adopt a tolerant attitude towards patient/clients, co-workers or professional association (the Polish Chamber of the Physiotherapists) authorities (§1.1, 2, 3, 4; §2.1, 9, 11, 12; §4.1; §10.3; §11.1). Those fragments stress particularly the meaning of respect towards law (including human and patient rights), professional regulations, autonomy, dignity, privacy, intimacy and physical integrity.

### 4.7. Dignity

Dignity is mentioned in the Ethical Code of Conduct for Physiotherapists together with respect and autonomy (§1.1). Such grouping seems justified, as dignity cannot be realised without respect. If someone is treated with dignity, he is also treated with respect. If, on the other hand, someone is treated with respect, his dignity is also preserved. The relation between dignity and autonomy is slightly different. Autonomy can be spoken about only in the case of competent individuals (self-aware and rational), while dignity (understood as human dignity) is a quality belonging to all humans. Therefore, both competent (autonomous) and incompetent individuals should be treated with dignity.

Dignity is a complex value and can be understood in at least three ways. (i) Human dignity is ascribed to all representatives of the human species, independently of the stage of development, age, sex, race, health status, social status, etc. It is sufficient to belong to the human species to deserve to be treated with dignity. (ii) Personal dignity, on the other hand, applies to the system of values of a conscious individual. The system of values makes an individual person place the net of values on the experienced world, thus giving this world proper sense and meaning. Unless the system of values is harmful to other people (or to other living creatures), an individual professing it should be treated with appropriate respect. (iii) Dignity can also be related to social (professional) roles undertaken by an individual. A person deserves respect for the very fact of being a physiotherapist, a doctor, a teacher, a shopping assistant, etc. Hence, nobody should be treated with lack of equality for the sake of the social role he fulfils [23].

Physiotherapists in their professional practice encounter the notion of dignity in its full understanding. This value relates directly not only to the good of a patient/client, but also to the good of the co-workers and the profession. Although dignity has been mentioned in the analysed document only four times (§1.1, 3; §2.1; §4.5), it can be deduced from nearly each and every ethical rule, as it co-creates the ontical status of physiotherapy.

### 4.8. Autonomy

Autonomy has been mentioned three times in the Ethical Code of Conduct for Physiotherapists (§1.1, 4; §2.1). In §1.1, autonomy has been quoted together with respect and dignity as the axiological basis of the profession. In §1.4 and §2.1, in turn, autonomy has been mentioned as the basic value regulating the relation between a physiotherapist and a patient/client.

Autonomy should be analysed, first of all, from the perspective of a patient/client and a physiotherapist. It should, however, be remembered that these values concern only competent individuals (self-aware and rational) who are able to take rational decisions and accept responsibility for them. The relation between a physiotherapist and a patient/client starts by expressing a consent to physiotherapeutic service by the latter [24]. As early as at this stage, a physiotherapist should develop partner relations with a patient/client, provide him or her with clear and reliable information, ensure privacy, intimacy and preserve professional secrets. Such an attitude on the part of a physiotherapist requires that a patient/client also engages in the relation actively, controls its course, offers his or her own opinions, or even resigns from the service, regardless of a physiotherapist’s opinion.

Nevertheless, physiotherapists should also be guaranteed autonomy while performing their services. It allows them to approach their profession creatively, adapting only to the needs of a patient/client and possibilities of their realisation (substantive, formal or technical). Physiotherapists possessing autonomy in undertaking professional decisions adopts responsibility for their realisation as well. Autonomy is especially important in multidisciplinary teams [25]. In these teams, each professional should function within the boundaries of his or her professional competence; hence, there should be no interfering into each other’s duties and competences. In such a situation, there would be no problem with defining a person responsible for certain activities [3]. The lack of autonomy, in turn, translates into low self-esteem of physiotherapists, heightened stress at work and getting professionally burnt out faster. That, in turn, results in a lower quality of the services they offer [3,26].

## 5. Conclusions

A code of ethics is necessary in jobs whose object is constituted by values particularly important for man and society. In physiotherapy, this is ability, complementing such values as health and life. Documents regulating the work of medical practitioners are necessary from the moral point of view for the professionals to exist and function, but also to fully understand their status in health care by patients/clients. Ethical rules should be a carrier of values reflecting social changes. Hence ethical documents should be continuously modified and supplemented with new phenomena and challenges reflecting the dynamics of social changes.

The Ethical Code of Conduct for Physiotherapists inscribes well in the global standards concerning the moral functioning of physiotherapists. The analysed values reflect standards that have roots in the ancient principle primum non nocere. Currently, values constituting the axiological basis of physiotherapy are care, professionalism, responsibility, justice, professional integrity, respect for dignity and autonomy of a patient/client. The undertaken analysis has shown those values constitute a dynamic system, in which they constantly come into relations, interactions and dependencies. Therefore, those values could adopt different meanings depending on the context in which they are realised.

Each physiotherapist should make an effort in order to reach values contained in the code of ethics and regularly confront them with his knowledge and professional experience. Physiotherapists should internalise those values and make them a signpost of their professional development. The process of the internalisation of values should start already during training and continue throughout the professional life of a physiotherapist. Those values should enrich the work of physiotherapists and make them reflect on their knowledge and professional competences, while being simply good human beings in the eyes of other people.

So far, no comprehensive study has been conducted concerning Polish physiotherapists’ knowledge of ethical documents regulating that profession. It might result from the fact that the current formal legal status of the profession was defined in 2015 and the Ethical Code of Conduct for Physiotherapists was adopted as late as in 2022. However, we find conducting such research necessary, as it would provide valuable feedback on the practical functioning of the introduced ethical regulations. It would also be worth introducing training sessions for physiotherapists, during which they could not only become familiar with the ethical documents currently in force, but first of all have an opportunity to undertake axiological analysis and reflection on their own professional practice. We hope that our article will be an important voice initiating actions leading in that direction.

## Data Availability

The data presented in this study are available on request from the corresponding author. The data are not publicly available due to privacy restrictions.

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
