# Peer review of "The Ethical Code of Conduct for Physiotherapists—An Axiological Analysis"

_ijerph, 2023, doi:10.3390/ijerph20021362_

Round 1

Reviewer 1 Report

Thank you for your work on this very important topic area. I enjoyed reading your paper and I believe, with clarifying edits, it will contribute to advancing the knowledge on The Ethical Code of Conduct for Physiotherapists in Poland. 

Suggestions/comments:

The Ethical Code of Conduct for Physiotherapists aims to educate physical therapists, students, other health care professionals, regulators, and the public regarding the core values, ethical principles, and standards that guide the professional conduct of the physical therapist. 

Consider including  paragraphs mentioning:

a) What is the level of knowledge among Polish physiotherapists of the ethical documents which constitute a comprehensive axiological basis of physiotherapy in Poland ? 

b)  What are the current educational/training opportunities ( if any) for practicing physiotherapists for staying up to date with the current code of conduct ? 

c) Are there any other ethical training requirements for this profession?

Author Response

Thank you for the critical review of the article. We fully agree with the suggested changes. We added to the article a fragment (marked in green) in which we suggested a need for research on physiotherapists’ knowledge of ethical documents. We also made a suggestion of the possibility of organising  training sessions on ethical regulations, enabling, among others, undertaking axiological analysis/ reflection on professional practice. We hope that the changes suggested by us will substantively complement the article.

Reviewer 2 Report

Very well researched. Interesting topic which deserves to be published. I do suggest that the shift between "patient" and "client" might need further explanation and was completed in different time periods in different nations. Different countries refer to "patients" or to "clients" depending on how actively they pursue models of patient empowerment and patient rights. I think some additional international information would add a great deal to the article. Good work.

Author Response

Thank you for the critical review of the article. We fully agree with the suggested changes. We added to the article a fragment (marked in yellow) in which we explained the difference between the relations between a therapist - a patient and  a therapist - a client in the physiotherapeutic practice. We added as well the differentiation introduced by Jaque Lacan, who defined a third pair in relations, this time between the analyst and the analysand, being an intermediate state between the two pairs in relations mentioned before. We also  introduced three additional items in the bibliography. We hope that the changes suggested by us will substantively complement the article.
